# Mathematical modelling to estimate the impact of maternal and perinatal healthcare services and interventions on health in sub-Saharan Africa: A scoping review

**Joseph H. Collins**⦿*, **Valentina Cambiano, Andrew N. Phillips, Tim Colbourn**

Institute for Global Health, University College London, London, United Kingdom

* joseph.h.collins@ucl.ac.uk

**Data Availability Statement:** All relevant data are within the manuscript and its Supporting Information files.

## Abstract

### Introduction

Mathematical modelling is a commonly utilised tool to predict the impact of policy on health outcomes globally. Given the persistently high levels of maternal and perinatal morbidity and mortality in sub-Saharan Africa, mathematical modelling is a potentially valuable tool to guide strategic planning for health and improve outcomes.

### Methods

The aim of this scoping review was to explore the characteristics of mathematical models and modelling studies evaluating the impact of maternal and/or perinatal healthcare interventions or services on health-related outcomes in the region. A search across three databases was conducted on 2nd November 2023 which returned 8660 potentially relevant studies, from which 60 were included in the final review. Characteristics of these studies, the interventions which were evaluated, the models utilised, and the analyses conducted were extracted and summarised.

### Results

Findings suggest that the popularity of modelling within this field is increasing over time with most studies published after 2015 and that population-based, deterministic, linear models were most frequently utilised, with the Lives Saved Tool being applied in over half of the reviewed studies (n = 34, 57%). Much less frequently (n = 6) models utilising system-thinking approaches, such as individual-based modelling or systems dynamics modelling, were developed and applied. Models were most applied to estimate the impact of interventions or services on maternal mortality (n = 34, 57%) or neonatal mortality outcomes (n = 39, 65%) with maternal morbidity (n = 4, 7%) and neonatal morbidity (n = 6, 10%) outcomes and stillbirth reported on much less often (n = 14, 23%).

**Funding:** JHC, AP and TC were funded by the UK Research and Innovation Global Challenges Research Fund (Grant reference MR/P028004/1, https://www.ukri.org/) followed by the Wellcome Trust (Grant reference 223120/Z/21/Z, https://wellcome.org/) during the inception and completion of this study as part of the Thanzi La Onse and Thanzi La Mawa projects. The funders had no role in study design, data collection and analysis, decision to publish, or preparation of the manuscript.

**Competing interests:** The authors have declared that no competing interests exist.

## Discussion

Going forward, given that healthcare delivery systems have long been identified as complex adaptive systems, modellers may consider the advantages of applying systems-thinking approaches to evaluate the impact of maternal and perinatal health policy. Such approaches allow for a more realistic and explicit representation of the systems- and individual- level factors which impact the effectiveness of interventions delivered within health systems.

## Introduction

The evaluation of the impact of healthcare programmes or policies on population health is a central component of public health, epidemiological and health economics science. Whilst the effects of biomedical interventions, such as drugs, on health outcomes are almost exclusively evaluated using randomised control trials (RCTs), experimental measurement of the direct effect of health policy interventions (e.g., changes to service coverage), may not be feasible due to ethical or practical constraints [1]. In such instances, mathematical models, defined here as models which "set out a theoretical framework that represent the causal pathways and mechanisms linking exposures, interventions, and infection or disease", can be utilised to estimate policy intervention effectiveness [1]. Types of mathematical models applied within health research vary considerably according to intended application, including simple compartment models, such as the Susceptible Infected Recovered (SIR) model, and dynamic simulation models such as systems dynamics or agent-based models in which complex systems are simulated over time [2,3]. Whilst the structure, assumptions and relative merits or limitation of such models varies considerably [1], the inherent flexibility of mathematical modelling, coupled with other intrinsic benefits such as lower relative cost when compared to experimentation, and the ability to synthesise multiple different data sources, has meant that they have come to be used extensively in the development of health policy at both the local governmental [4] and the global [5] levels.

Importantly, the application of mathematical models to inform health policy has the potential to generate significant health gains, particularly within the field of global maternal and newborn health. Evidence suggests that, despite significant progress since the turn of the century [6,7], maternal and perinatal mortality remains unacceptably high, particularly in regions such as southeast Asia and sub-Saharan Africa (SSA) [8–10], due in part to insufficient coverage and quality of key maternity services [11–16]. As such, there are a substantial number of pertinent policy questions regarding how best to deliver high-quality maternity services to mothers and newborns to reduce mortality and maximise health which are of particular interest to policy makers within the region. This is especially true given the prominence of maternal and newborn health metrics within the health-related Sustainable Development Goals targets [17]. For example, significant attention is now being paid to the potential health gains associated with maternity service delivery redesign [18,19], defined as the "intentional reorganisation of a health system to improve equity, quality, and outcomes" [19] which entails redesigning maternity services so that births occur exclusively within higher level or advanced facilities in line with practice in most high-income countries [18]. Evaluation of such policies through experimentation, such as through a cluster RCT, is associated with substantial ethical constraints such as a potential lack of clinical equipoise. Additionally, certain outcomes of interest, such as maternal mortality, may be sufficiently rare that the required sample size is prohibitive to conducting a trial. In such circumstances, mathematical modelling may offer

important insights into the potential impact of alternative configurations of intervention or service delivery.

Whilst there is evidence of the growing use of modelling approaches within public health more generally [2], to date there has been no systematic evaluation of the broad application of mathematical modelling to evaluate maternal and newborn services within the region of SSA. The review presented here has therefore been conducted to answer the primary research question of: *"What are the characteristics of mathematical models which have been applied within the academic literature to estimate the impact of the delivery of maternal and/or perinatal healthcare interventions or services on health-related outcomes in sub-Saharan Africa?"*. To best answer this question, the review has three primary objectives: (i) to characterise and map the relevant literature base; (ii) to explore the characteristics of the models and modelling studies found within this literature and (iii) to critically discuss the most common modelling approaches utilised. Systematic evaluation of the relevant literature allows for identification of common approaches taken, trends in study and model characteristics and potential gaps within the literature. Due to the broad scope of the research question and inclusion criteria we considered it appropriate to conduct a scoping review as opposed to a systematic review.

## Materials and methods

Levac et al's. [20] framework was chosen to guide the conduct of this review as it provided the most comprehensive and clear guidance building upon the original six-stage framework for scoping review conduct developed by Arksey and O'Malley [21]. This framework was utilised in tandem to the PRISMA-ScR checklist developed by the Joanna Briggs institute [22] to ensure the content of the review is in keeping with recommended best practice. The checklist for this review is provided in the (S1 File). We did not previously develop a review protocol.

### Inclusion criteria

Studies were eligible for inclusion if they described analyses using a mathematical model, as defined in the introduction, which reported the potential effect of the implementation or change in coverage of a maternal or perinatal health intervention on maternal or perinatal epidemiological or health related outcomes.

Maternal and perinatal healthcare interventions or services (referred to as interventions for the remainder of this review) are those interventions delivered as part of antenatal, intrapartum, or postpartum care to women at any stage along the pregnancy continuum (i.e., from conception to 42 days postnatal) or their newborns (i.e., from birth to day 28). For mothers, we only consider interventions intended to prevent morbidity or mortality associated with conditions or complications directly caused by the processes of pregnancy or birth.

Health outcomes include any measure of morbidity or mortality within the target population. This is not limited to the primary outcomes of the study and any studies which report outcomes partially derived from other health outcomes (e.g., Incremental Cost Effectiveness Ratio (ICER)) or that report health outcomes as secondary outcomes were included.

The geographical area of focus for this review was SSA with the full list of included countries being adapted from the UN Statistics Division geographic regions [23]. Whilst the term SSA is a largely arbitrary categorisation of many extremely heterogenous and diverse countries, it was deemed a logical regional category as this grouping of African nations has been used extensively within the global health literature and therefore utilised to ensure the maximum number of relevant papers were returned.

Finally, papers in which mathematical models were described but not applied were not included, however those which used models previously applied in were. In addition, only

English language papers were included in the review due to unavailable resources to undertake translation and/or non-English language searches.

## Search strategy

A multi-stage search strategy was employed to identify relevant studies for inclusion. First, a systematic search of three electronic databases was undertaken on 2nd November 2023. The databases included PubMed, Web of Science and Global Health (1973- present). Once relevant studies were identified, a hand-search of the bibliographies of selected studies was undertaken to identify any additional relevant sources of information. Table 1 presents the search terms used for searching the electronic databases. Due to the proliferation of studies identified in the first stage in which analyses were conducted using the Lives Saved Tool (LiST) [24], we undertook an additional search using the PubMed database using the search term "lives saved tool" to ensure all relevant studies were included in the final sample.

Citations returned from each database search were imported into Mendeley Desktop citation manager (v1.19.8) and pooled. Using in-built functionality automatic deduplication was undertaken followed by a manual review of returned studies to ensure any duplicates missed by the automatic search were removed.

## Study selection

A three-step process was employed to identify relevant studies from retrieved titles; 1.) the titles of the total number of retrieved citations were screened for apparent relevance using the inclusion criteria, 2.) the potentially relevant studies were separated then screened by the contents of their abstract against the eligibility criteria, 3.) those which still appeared relevant were screened through review of the full text and included in the review if appropriate. This process was mirrored with any studies identified via the hand-search of the reference lists of the final studies selected from the database and those identified through the targeted search. Despite recommendations within current methodological frameworks that this process, and the

**Table 1. Search terms used in database search of scoping review.**

| Key term/concept | Search Terms |
|---|---|
| *Mathematical Model(s)* | ("mathematical" OR "disease" OR "microsimulation" OR "deterministic" OR "stochastic" OR "compartment*" OR "individual-based" OR "individual based" OR "agent-based" OR "agent based" OR "discrete-event" OR "time to event" OR "system dynamics" OR "monte carlo" OR "markov chain" OR "simulation") AND ("model*" OR "simulat*") |
| *Maternal and/or Perinatal Health* | "matern*" OR "pregnan*" OR "obstetric*" OR "antenatal care" OR "prenatal care" OR "postnatal care" OR "postpartum care" OR "birth attendan*" OR "intrapartum care" OR "labour and delivery" OR "facility delivery" OR "child birth" OR "childbirth" OR "neonat*" OR "perinatal" OR "newborn" OR "new born" OR "BEmONC" OR "EmONC" OR "CEmONC" OR "emergency obstetric and newborn care" OR "emergency obstetric and new born care" |
| *Sub-Saharan Africa* | Angola* OR Benin OR Botswana* OR "British Indian Ocean Territory" OR "Burkina Faso" OR Burundi* OR "Cabo Verde" OR Cameroon* OR "Central African Republic" OR Chad OR Congo OR Comoros OR "Cote d'ivoire" OR "Democratic Republic of the Congo" OR Djibouti* OR Eswatini* OR Eritrea* OR Ethiopia* OR "equatorial guinea" OR "French Southern Territories" OR Gabon OR Gambia* OR Ghana* OR guinea OR "Guinea-Bissau" OR Kenya* OR Lesotho OR Liberia* OR Madagascar* OR Malawi* OR Mali* OR Mauritania* OR Mauriti* OR Mayotte OR Mozambi* OR Namibia* OR Niger* OR Nigeria* OR "Saint Helena" OR Reunion OR Rwanda* OR "Sao tome and Principe" OR Senegal* OR Seychell* OR "Sierra Leon*" OR Somalia* OR "South Africa*" OR "South Sudan" OR Tanzania* OR Togo* OR Uganda* OR Zambia* OR Zimbabwe* OR" sub-Saharan Africa" |

process of extraction and analysis, should be conducted by a multidisciplinary team to improve the validity of the review's findings [20,22,25,26], this review was initially conducted as part of the first authors Ph.D. and therefore was conducted with no additional authors.

### Data extraction and analysis

Following the selection of relevant studies, data was abstracted from the final studies via a data abstraction form. The development and use of a unique abstraction form is central to scoping review methodology and allows researchers to identify and extract all pertinent information from a given study which relates directly to the reviews primary research question [20,27]. As such a preliminary data extraction form was first developed and trialled on a subset of five studies, in-keeping with recommended scoping review practice [20,27], however due to the significant heterogeneity in study methods and analyses a second round of trialling was conducted on a further five studies after the initial refinement of the data extraction form. Within the final form 35 unique data items are identified. They can broadly be categorised as general characteristics of the study (8 items), characteristics of modelled interventions (9 items), characteristics of the mathematical model (7 items) and characteristics of any conducted analyses (11 items). The final completed data extraction form can be found in the (S2 File).

Data items relating to model characteristics were adapted from Garnett et al. [1], where the authors provide defining criteria relevant across different mathematical modelling methodologies. These criteria, which have been refined for data extraction, include how feedback between interventions and outcomes are represented in the modelled system, the role chance plays in model behaviour, how individuals are represented within the model and how variable change is governed with regards to time [1]. These criteria are briefly summarised in Fig 1.

Following data extraction, data items were synthesised quantitatively where appropriate (i.e., in the presentation of percentages and frequencies) and summarised under relevant subheadings presented in the following section.

## Results

### Search results

A total of 10,980 potentially relevant studies were identified, which was reduced to 8,660 after the removal of duplicates. Following title screening, the abstracts of 150 papers were screened (including all papers from hand-searching) using the inclusion criteria followed by full-text screening of 90 most relevant. Of these studies a further 30 were excluded leaving a final total of 60 papers for review [28–87]. Fig 2 provides a diagrammatic representation of the search and screening processes within this review.

### General study characteristics

Table 2 details the general characteristics of the included studies. The publication date of the included studies skewed towards the latter half of the 2010s, with 70% of studies published after 2015 (n = 42) and the greatest number of studies being published in 2019 (n = 9 (15%)). In over half of studies (n = 33, 55%) the modelled 'population' was representative of a single country, whilst the remaining studies either modelled interventions applied to populations at a regional level, such as sub-Saharan Africa (n = 8), or to two or more countries or regions (n = 19).

Location of the affiliated institution of the first study author was extracted, alongside whether any co-authors were based in institutions within study countries. Seventy percent (n = 40) of studies' first authors institutions were not located in any of the study countries with

| Feedback within model system | |
|---|---|
| **Linear -** The modelled link between an exposure or treatment and outcome (e.g. disease) is expressed as a linear function.<br><br>*For example, if a given treatment halves the risk of disease acquisition, and that treatment is delivered to everyone at risk of the condition, then the incidence of that condition would reduce by half in the population.* | **Non-linear** - The modelled link between an exposure or treatment and outcome (e.g. disease) expressed as a non-linear function.<br><br>*For example, risk of infectious disease acquisition is modelled to be determined by rate of contact between susceptible and infected individuals, the number of infected already within the population, rate of transmission etc.* |

| Role of chance | |
|---|---|
| **Deterministic -** Behaviour of the model, for a defined population, is fixed. Multiple runs of the model with the same parameter set leads to the same result. | **Stochastic -** Behaviour of the model incorporates randomness.<br><br>*For example, events occur to an individual or population group within the model according to random draws against a probability parameter.* |

| Representation of individuals | |
|---|---|
| **Individual level -** The model follows a set of distinct individuals over time, with each individual being 'known' | **Population level-** The model follows a group within the population without the specification of which individuals are involved. |

| Timestep | |
|---|---|
| **Age -** Events within the model are experienced by individuals as they age.<br><br>*For example, a model of a single birth cohort.* | **Calander** - Events within the model are experienced following the progression of time. |

*Figure contents adapted from Garnett et al. (1)*

**Fig 1. Overview of key model characteristics.**

nearly half of total studies led by a first author based at an institution in the United States (n = 29, 48%) and only seventeen studies (28%) were led by first authors based at institutions within SSA. As evident in Table 2, approximately half of studies (n = 29) had no authors based at an affiliated institution within any of the study countries.

Finally, the primary aim of included studies was evaluated. Whilst all studies reported at least one outcome associated with either maternal or perinatal health in line with inclusion criteria, over a third (n = 22, (37%)) aimed to estimate costs associated with modelled interventions or calculate the interventions' cost-effectiveness.

**Intervention characteristics.** Table 3 summarises extracted data relating to the interventions evaluated within the reviewed studies. Interventions were extracted, as written, from either within the main text or supplementary material if indicated by authors. They were then alphabetised to assess frequency and, where wording differed slightly between interventions which could reasonably be considered the same, this was corrected.

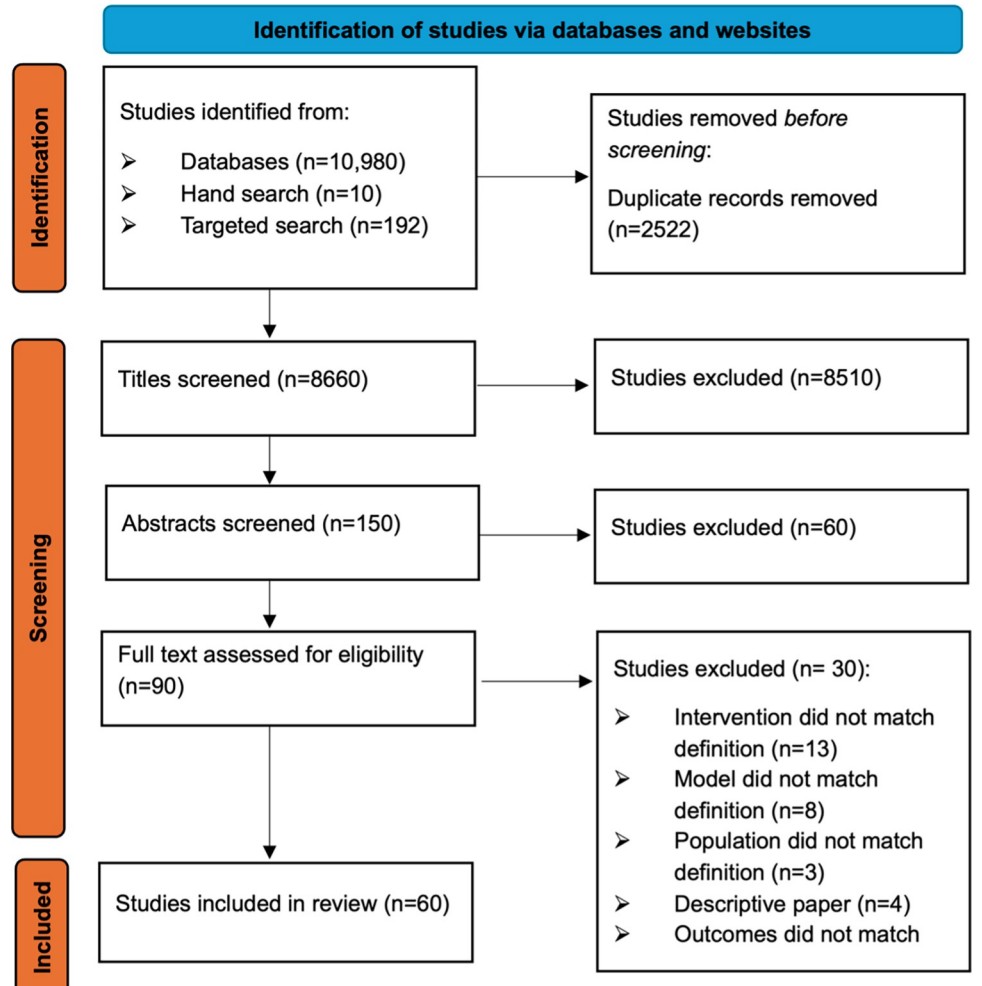

**Fig 2. PRISMA diagram reporting search results.**

There was extensive heterogeneity between studies in relation to the interventions which were evaluated by mathematical models and the ways in which interventions were defined. Across the 60 included studies a total of 1055 interventions (inclusive of duplicates) were described, with most studies (n = 52 (87%)) evaluating the impact of multiple different interventions and fewer studies evaluating single interventions. This value includes all modelled interventions including those which are delivered outside routine antenatal, intrapartum, or postpartum care or represented nonclinical interventions such as improved access to clean water.

Commonly, studies evaluated scale-up of interventions which could reasonably be defined as part of standard clinical practice in most settings (e.g., basic newborn resuscitation). Much less frequently, (n = 7, 12%) studies evaluated novel interventions which are not routinely available to mothers or newborns in most settings. This includes studies in which the aim was to predict the effect of an intervention on health outcomes that acts through improving coverage of other key maternity services. For example, Wilcox et al. [29] predicted the impact of a Mobile Technology for Community Health initiative in Ghana which has been shown to improve skilled birth attendance and facility delivery and modelled the effect of this intervention using LiST. Other examples include Carvalho et al. [56] in which the authors estimated

**Table 2. General characteristics of the reviewed studies.**

| General study characteristics | Studies (N = 60) | Percentage (%) |
|---|---|---|
| *Year of publication*: | | |
| Pre 2010 | **4** | *7%* |
| 2010–2014 | **14** | *23%* |
| 2015–2019 | **34** | *57%* |
| 2020–2023 | **8** | *13%* |
| *Single or multiple countries modelled*: | | |
| Single | **33** | *55%* |
| Multiple | **27** | *45%* |
| *Most modelled countries (top 5)* [a] | | |
| Ethiopia | **21** | *35%* |
| Tanzania | **19** | *32%* |
| Uganda | **18** | *30%* |
| Kenya | **18** | *30%* |
| Malawi | **16** | *27%* |
| *In addition to health outcomes the study aimed to evaluate cost/cost-effectiveness*: | | |
| Yes | **22** | *37%* |
| No | **38** | *63%* |
| *First author affiliation*: | | |
| Within study country | **18** | *30%* |
| Outside study country | **42** | *70%* |
| *Any author based at institution within the/a study country*: | | |
| Yes | **31** | *52%* |
| No | **29** | *48%* |

[a]Sourced from studies which explicitly listed modelled countries. Additionally, multiple countries are modelled in many papers therefore values in this row when summed exceed the total number of papers.

the cost-effectiveness and health impacts of an inhaled oxytocin product and Ali et al. [59] who evaluated the Augmented Infant Resuscitator, a novel device which provides real-time feedback on ventilation quality during the resuscitation of newborns.

Almost all studies evaluated at least one intervention which was delivered to women (N = 57 (95%)), even including those in which maternal outcomes were not evaluated or reported (e.g., evaluating the effect of maternal nutritional supplementation in ANC on newborn outcomes). As shown in Table 3 these interventions were most frequently delivered during the antenatal and intrapartum periods of pregnancy whilst postpartum interventions delivered to mothers were less frequent (n = 41, 68%). The evaluation of interventions delivered to neonates occurred slightly more frequently and was presented in over two-thirds of studies (n = 41 (68%)). Of the studies in which only neonatal interventions were evaluated (n = 2), Tsiachristas et al. [30] evaluated a task shifting intervention for routine nursing tasks delivered to neonates in a Kenyan neonatal intensive care unit (NICU) whilst Ali et al. [59] has been described above.

**Model characteristics.** Table 4 summarises the extracted data relating to the models applied within the reviewed studies. Approximately a quarter (n = 16 (27%)) of studies utilised a model which had been developed specifically for the purpose of achieving one or more of the study's aims whilst the remaining employed models which had been developed prior to study conduct. Over half of studies (n = 34, 57%) reported analyses conducted using LiST, and eight

**Table 3. Characteristics of the interventions included within the reviewed studies.**

| Intervention characteristics | Studies (N = 60)[a] | Percentage (%) |
|---|---|---|
| Study evaluated a single or multiple interventions: | | |
| *Single* | **8** | *13%* |
| *Multiple* | **52** | *87%* |
| Study modelled an intervention not currently delivered as part of routine service delivery: | | |
| *Yes* | **7** | *12%* |
| *No* | **53** | *88%* |
| Study includes interventions delivered to: | | |
| *Women* | **57** | *95%* |
| *Neonates* | **41** | *69%* |
| If a study modelled any interventions delivered to mothers, they were delivered: | | |
| *Antenatally* | **47** | *78%* |
| *During labour/birth* | **43** | *72%* |
| *Postnatally* | **41** | *68%* |
| Study modelled interventions targeting: | | |
| *Maternal health outcomes* | **42** | *70%* |
| *Neonatal health outcomes* | **46** | *77%* |
| *Stillbirths* | **17** | *28%* |
| Most evaluated interventions (Top 5) [b]: | | |
| *1. Immediate assessment and stimulation / Neonatal resuscitation* | **34** | 57% |
| *2. Skilled birth attendance   3. Vitamin A supplementation   4. Clean birth or postnatal practices   5. Magnesium Sulphate* | **32** | 53% |
| | **24** | 40% |
| | **24** | 40% |
| | **20** | 33% |

[a]The values in the some of the rows exceed the total number of papers when summed, as most studies modelled multiple interventions.

[b]Excluding those interventions which are not routinely delivered as part of antenatal, intrapartum, or postpartum care.

studies (13%) using the Maternal and Neonatal Directed Assessment of Technology model (MANDATE).

Of the remaining studies in the sample, most utilised a decision tree/analytic model (n = 7). These models, commonly used in health economic analysis, represent potential treatment pathways as a series of 'branches' demarcated by decision nodes, representing some decision of interest (e.g., the decision to deliver a screening intervention) and chance nodes from which mutually exclusive probabilities of a set of outcomes following a decision are represented [88,89]. Other types of models were utilised much less frequently with only 5 (8%) studies employing an individual-based framework [28,33,35,69,87].

Model characteristics were extracted from all included studies guided in part by Garnett et al. [1] (Fig 1). Considering whether the link between a given intervention and the health outcome in the model is represented as a linear or non-linear function, 95% of models can be described as linear (n = 57). A simple linear model would assume that if a given treatment halves the risk of disease acquisition, and that treatment is delivered to everyone at risk of the condition, then the incidence of that condition would reduce by half in the population. Linear representations are common in cohort models where it is assumed that the outcome of different individuals represented in the model is independent (i.e., treatment delivered or not delivered within the model to one individual does not affect the treatment or outcomes of others). The fact that most models reviewed here are linear is unsurprising due to the number of studies using either LiST or the MANDATE model which both are self-described as linear. Alternatively, models which represent this link as non-linear functions were much less common (n = 2) [32,35].

Next, considering the incorporation of chance into the modelling framework, few models (n = 5, 8%) were found to be stochastic (i.e., events occur to an individual or population group

**Table 4. Characteristics of the models and analyses performed within the reviewed studies.**

| Characteristics of the models and analyses performed | Studies (N = 59) | Percentage (%) |
|---|---|---|
| Was the model developed to conduct this study: | | |
| Yes | 16 | 27% |
| No | 44 | 73% |
| Model employed by study authors: | | |
| LiST | 34 | 57% |
| Maternal and Neonatal Directed Assessment of Technology model (MANDATE) | 8 | 13% |
| Decision Tree/Analytic model | 7 | 12% |
| Individual or Agent-based model | 5 | 8% |
| Systems-dynamics model | 1 | 2% |
| Other/Undefined | 5 | 8% |
| Model structure, use or data and reporting: | | |
| System feedback–Linear | 58 | 97% |
| System feedback–Non-linear | 2 | 3% |
| Deterministic | 55 | 92% |
| Stochastic | 5 | 8% |
| Population/cohort based. | 55 | 92% |
| Individual-based | 5 | 8% |
| Time represented as calendar time. | 53 | 88% |
| Time represented with age | 7 | 12% |
| Diagrammatic representation in manuscript | 13 | 22% |
| No diagrammatic representation in manuscript | 47 | 78% |
| (Where relevant (n = 53)) Calendar-time horizon for analysis was: | | |
| One year or less | 20 | 34% |
| 2–5 years | 11 | 18% |
| 6–10 years | 5 | 8% |
| 11+ years | 17 | 28% |
| (Where relevant (n = 7)) Age-time horizon for analysis was: | 3 | 5% |
| A delivery | 1 | 2% |
| A pregnancy and delivery | 1 | 2% |
| 2 years | 2 | 3% |
| Lifetime | | |
| Sensitivity analysis was conducted and reported on: | | |
| Yes | 20 | 33% |
| No | 40 | 67% |
| Study reported outcome relating to: | | |
| Maternal mortality | 34 | 57% |
| Maternal morbidity | 4 | 7% |
| Maternal DALYs | 6 | 10% |
| Neonatal mortality | 39 | 65% |
| Neonatal morbidity | 6 | 10% |
| Neonatal DALYs | 7 | 12% |
| Stillbirths | 14 | 23% |
| For multi-country studies (n = 27) results were reported by country or as an aggregate over a region/group? | 9 | 33% |
| Country-specific | 18 | 67% |
| Aggregate | | |

within the model according to random draws against a probability parameter), while most utilised a deterministic approach to derive model results, meaning that multiple model runs with the same parameter set will always return the same result (n = 55). All stochastic models within the review employed an individual-based approach, where each member of the population is modelled individually with distinct characteristics which can be tracked, whilst the remaining models included representation of population level processes and trends using input parameters representing averages and leading to population-level results.

Description of model structure and components varied according to model type. Only 13 (22%) papers presented any diagrammatic representation of the model's underlying structure within the manuscript or supplementary material. However, of the papers without a model diagram over 80% of these papers utilised either LiST or MANDATE model. Diagrammatic representation of model structure is available for both these models on their associated websites [90,91]. This meant that papers in which novel models were reported were significantly more likely to include a diagram of the structure.

**Study analyses and outcomes.** Table 4 also provides broad overview of the analyses and relevant outcomes extracted from the reviewed studies. All study outcomes were extracted, as presented, from each study and then sorted according to categories presented in Table 4. Of note, cost related outcomes are not included in the table but are available in the (S2 File). Maternal and neonatal mortality outcomes were by far the most frequently reported within the sample with very few studies reporting outcomes which could reasonably be classified as relating to maternal (n = 4, 7%) or neonatal (n = 6, 10%) morbidity. Interestingly, of the studies presenting maternal morbidity outcomes [39,46,56,62] all reported some measure of postpartum haemorrhage incidence or prevalence whilst newborn morbidity outcomes ranged from the prevalence of stunting or wasting [57] to the number of cases of intrapartum related neonatal events and injury [44]. Additionally, any measure of the number or rate of stillbirths within a modelled population was reported in only around a fourth of studies within the sample (n = 14, 23%).

Additionally, we considered how outcomes were presented for studies in which more than one country population was theoretically modelled. Surprisingly, of the multi-country studies (n = 27), only eight reported results from their analysis disaggregated by each included country, with the rest presenting the results for a group of countries or even an entire region, such as SSA. Whilst for some studies this was an analytical choice, for those using the MANDATE model authors were limited by the design of the model which currently only represents the region of SSA or the country of India within the model's base assumptions (e.g., condition incidence, intervention availability).

## Discussion

This scoping review has provided a comprehensive and descriptive overview of mathematical modelling within maternal and perinatal health research within SSA. Through application of Levac et al's. [20] framework and guided by the recent PRISMA-ScR checklist [22], a replicable evaluation of the published academic literature has been undertaken in which the characteristics of relevant mathematical models and their application in the evaluation of maternal and/or perinatal interventions has been summarised. Given that the number of studies in the field utilising these approaches appears to have increased over time, as evidenced in this review, this is a timely and pertinent overview of commonly utilised modelling approaches. The proliferation of studies from 2015 onwards could be attributed to several factors including the utility of such methods in evaluating pathways towards achieving maternal and perinatal health related goals within the Millennium Development Goals (MDGs) and Sustainable Development Goals (SDGs), as demonstrated in several reviewed studies [80–86].

Our review found that the most often utilised model within this field was LiST, which was used in nearly sixty percent of the sample. LiST has long been recognised as a popular tool which has been utilised extensively within the field of global maternal and child health by a varied group of interdisciplinary stakeholders [92]. LiST is a linear deterministic model as it "describes fixed relationships between inputs and outputs, and the tool will produce the same outputs each time the model is run with identical inputs" [93]. The primary input is the

coverage of an intervention being evaluated and the primary outputs are either change in the cause-specific mortality or one of the included "risk factors" for mortality such as stunting [93]. LiST is extremely comprehensive in both the breadth of causes of maternal and child ill-health which are represented and the range of in-built interventions which modellers can manipulate within the tool and has been utilised in the evaluation of health programmes and to inform strategic planning at the national and local level with some organisations regularly using the tool and embedding LiST into organisational workflows by developing 'in-house' capacity to work with the model [92]. Importantly, as demonstrated in Table 4, many of the other models in this sample share key characteristics with LIST in that they are linear, deterministic, and changes in the modelled system are expressed at the population level.

However, healthcare delivery systems have long been identified as complex adaptive systems which exhibit explicitly non-linear behaviour by definition [94–96]. Variation in the coverage of established interventions, or integration of new interventions, may have unexpected down-stream effects on the outcome of interest despite evidence that the intervention is effective [97,98] which cannot be predicted using LiST or similar linear models. Within maternal and perinatal health this has been demonstrated in several Low- and middle-income countries (LMICs) in which, driven by political commitment to achieving the maternal health indicators within MDG 5 and SDG 3, the proportion of mothers who give birth within a health facility has increased significantly [97,98]. However, this has not been accompanied by an observed population-level reduction in maternal mortality, despite strong evidence for the efficacy of interventions delivered to mothers during and following labour [97,98]. The relationship between facility delivery coverage and mortality is complex, non-linear, and likely influenced by several individual and health system-level factors relating to quality of care such as the effect of increased volume of patient attendance and resource use [97,98]. Failure to understand the system environment and behaviours is often a driving factor in the failed implementation of new interventions, which may have been effective had a systems-thinking lens, one which considers the potential effect of system level phenomena on intervention or service implementation, been employed [99–101]. This issue is further compounded when the complexity of the interventions is increased, as many health systems strengthening interventions are largely programmatic service changes which mean current evaluation methodologies may be insufficient to produce valid or reliable outcomes [101].

Evaluation of relevant health processes at the population level, as taken by most models in this review, is advantageous in that it is computationally cheap and, in many circumstances, may be a sufficient level of detail to ascertain a reliable result [102]. However, in recent years, the use of individual-based model (IBMs) has become more popular within epidemiology and public health [2,103,104]. Broadly, IBMs are a subset of computational simulation models in which the model is designed to replicate the characteristics, behaviours, and actions of a set of 'agents', within a given environment, to reproduce complex population-level or systems phenomena [105–108]. Within epidemiological modelling, these 'agents' are often representative of individual people who may interact with one another to simulate a hypothetical population of interest, characterised by individual variables, and governed by sets of predetermined rules [106].

One of the primary strengths of this approach in the modelling of health processes is the ability to explicitly model individual-level variation and interactions between said individuals within the population [105]. Within epidemiological modelling, this approach has been employed in the modelling of infectious diseases, in which interactions between individuals allows for representation of complex transmission dynamics within the population [104]. This process, often referred to as 'interference', occurs because individuals within the model are not represented as independent, and therefore the assignment of an exposure, intervention or

treatment can be influenced by outcomes of other individuals in the model [106]. Interference cannot be represented in population-level models where the assumption of non-independence is not met which means such models are less suitable to explore the complexities of healthcare delivery. This is demonstrated in the study by Nadkarni et al. [35], who explicitly represent the relationship between the acuity of a group of patients in a maternity ward, the associated demand on consumables and, in-turn, the effect on outcomes.

Our review found that only 27% of research teams developed novel mathematical models to answer the research question of interest. Whilst the appeal of using a previously developed model is clear, the importance of considering country-context within the evaluation of health policy has long been identified within the literature, as national and regional context likely has significant impact on how interventions are implemented and accessed by populations [109]. Many modellers have embraced methods to embed contextual structures into model design through participatory approaches to model development [110], stakeholder workshops developing causal loop diagrams [111], or simply through explicitly representing the system of interest within a given context by supported model validation with key stakeholders [112]. Whilst such approaches were underutilised in the reviewed studies, context-informed modelling was well demonstrated by Semwanga et al. [32] who developed a novel systems-dynamics model to explore neonatal health interventions in Uganda through a process of stakeholder engagement and causal mapping with service users and policy makers.

Next, considering the commonly reported outcomes of modelling analyses, this review found that studies were considerably more likely to report outcomes relating to either maternal and neonatal mortality and much less likely to predict intervention effects on morbidity or stillbirths. Limited reporting on morbidity outcomes is not surprising due to the widespread use of LiST which does not calculate the effect of interventions on most morbidity outcomes. The absence of morbidity from the outcomes of modelling studies evaluating health intervention delivery is concerning considering that many of the routine interventions delivered to mothers and newborns are intended not only to prevent mortality but to reduce the incidence of common morbidity associated with pregnancy and the newborn period to improve overall quality of life [113–115]. The absence of such outcomes may mean that the true health impact of improving intervention coverage is significantly underestimated at the population level.

Finally, we found that very few of the included studies evaluated the potential impact of interventions on stillbirths. There are similarities between stillbirth research and morbidity research, with many arguing that stillbirth has historically been absent from the global maternal and perinatal health agenda for some time [116–118]. Qureshi et al. [118] suggest that stillbirths should be attributed the same 'value' as neonatal deaths in economic evaluation of maternal health interventions as they are commonly excluded in measurements such as life years gained or DALYs averted. This is problematic when considering very early preterm neonatal deaths are included in neonatal death statistics whilst a stillbirth at 41 weeks gestational age would not be, despite being more developmentally 'mature' [118]. Whilst recently a renewed focus on stillbirth prevention has led to considerable progress [119] a large number of stillbirths occur per year within SSA [8] and are associated with considerable morbidity in mothers [120]. Importantly several interventions to prevent stillbirth are well described in the literature through a Cochrane review of relevant trials [121] and through observational studies [122]. As such mathematical modelling approaches are well positioned to evaluate the context-specific effects of these interventions in high burden populations.

There are several important strengths and limitations of this review to acknowledge. The primary strength of this review is the use of a systematic search strategy to identify relevant studies and due to the nature and broad scope of the research question and associated objectives, the scoping review methodology was the most appropriate choice to map the relevant

literature. A traditional systematic review would not have been appropriate, as the aim of this review was not to synthesise the reported results of the modelling studies, something that would not have been possible due to the diverse set of methodologies employed within this research. To ensure that conduct of the review adhered to contemporary best-practice frameworks the PRISMA checklist [22] was used to guide the design and conduct of the review (S1 File).

Considering this review's limitations, firstly we have only included studies conducted in the English language, as this review was conducted as part of the first author's Ph.D., meaning potentially relevant studies reported in other languages were excluded. Secondly the search strategy did not include modelling analyses reported within the grey literature. Implicit exclusion of potentially relevant modelling analyses conducted in reports or government documentation could introduce bias into the review and potentially alter the review's conclusions [123]. Interestingly, the LiST team highlights in their most recent update supplement that many applications of the model lie outside of academic publishing, with efforts being made by the team to enhance the visibility of LiST applications by policy makers across LMICs [124]. Whilst frameworks for the systematic search of grey literature do exist, they do not alleviate common issues with comprehensive grey literature searching such as inbuilt search-engine filters which can introduce bias into source selection [125].

Thirdly, a significant number of modelling studies which evaluated solely the impact of health interventions on indirect causes of maternal morbidity/mortality were excluded. This was particularly true for modelling studies evaluating screening and treatment of HIV during pregnancy on HIV outcomes. Undoubtedly, widening the studies inclusion criteria so that such studies were reviewed would likely have changed the results of this review. This is particularly true given the proliferation of HIV models used to conduct research in SSA [126,127]. Finally, this review did not explicitly consider the quality of included studies by conducting a critical appraisal as part of the review process. Whilst evaluation of the quality of modelling studies conducted in this area is undoubtedly important, this process is outside of the remit of a scoping review [20,128] and is beyond the objectives of this review.

In conclusion, in this review we have reported on the methodology, conduct and results of a scoping review to explore the breadth of mathematical modelling research within evaluation of maternal and perinatal healthcare interventions in SSA. Results from this review indicate that whilst there is variation in the reviewed studies, the literature base is dominated by two models which, whilst distinct in their design, share commonalities in how they represent the relationship between intervention and outcome, populations of interest and the role randomness plays in the model system. Individual-based or systems-thinking approaches to mathematical modelling were much less commonly applied within the field, despite potential advantages such as a more realistic and explicit representation of the systems- and individual- level factors that impact the effectiveness of interventions delivered within complex-adaptive health systems.

## Supporting information

**S1 File. PRISMA-ScR checklist.**
(DOCX)

**S2 File. Data extraction form and extracted data.**
(DOCX)

## Author Contributions

**Conceptualization:** Joseph H. Collins, Valentina Cambiano, Andrew N. Phillips, Tim Colbourn.

**Data curation:** Joseph H. Collins.

**Formal analysis:** Joseph H. Collins.

**Methodology:** Joseph H. Collins, Valentina Cambiano, Andrew N. Phillips, Tim Colbourn.

**Supervision:** Valentina Cambiano, Andrew N. Phillips, Tim Colbourn.

**Writing – original draft:** Joseph H. Collins.

**Writing – review & editing:** Valentina Cambiano, Andrew N. Phillips, Tim Colbourn.

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
