## [Decision Letter · Decision Letter 0]

19 Jul 2024

PONE-D-23-41936Mathematical modelling to estimate the impact of maternal and perinatal healthcare services and interventions on health in sub-Saharan Africa: A scoping reviewPLOS ONE

Dear Dr. Collins,

Thank you for submitting your manuscript to PLOS ONE. After careful consideration, we feel that it has merit but does not fully meet PLOS ONE’s publication criteria as it currently stands. Therefore, we invite you to submit a revised version of the manuscript that addresses the points raised during the review process.

We look forward to receiving your revised manuscript.

Kind regards,

Ashish KC

Academic Editor

PLOS ONE

Additional Editor Comments:

Dear Dr. Collins

I have now received the reviewer's comment and I invite you now to review the comments and make necessary revisions with point to point response letter.

Best regards, Ashish

Reviewers' comments:

Reviewer's Responses to Questions

**Comments to the Author**

1. Is the manuscript technically sound, and do the data support the conclusions?

Reviewer #1: Yes

Reviewer #2: Yes

2. Has the statistical analysis been performed appropriately and rigorously? 

Reviewer #1: Yes

Reviewer #2: Yes

3. Have the authors made all data underlying the findings in their manuscript fully available?

Reviewer #1: Yes

Reviewer #2: Yes

4. Is the manuscript presented in an intelligible fashion and written in standard English?

Reviewer #1: Yes

Reviewer #2: Yes

5. Review Comments to the Author

Reviewer #1: Thank for inviting me to the interesting scoping review by Joesph and colleagues on the mathematical models uses to estimate the impact of maternal and perinatal health intervention in Sub-Saharan Africa". I read and reviewed the scoping review with interest as it describes the different modeling exercises done in the last 50 years to describe the impact of scaling up the intervention on perinatal mortality outcomes.

This is indeed an interesting paper, which provides what modeling were being done and which interventions were focused on.

Most of the modelling that were done were based on the different clinical or population-based studies done to assess the effectiveness of the intervention. What this paper has shown it that lot of modelling had been published between 2015-2019, indicating the work in the end of the MDG period and start of the SDG period. Most of the modeling were done for interventions before birth (antenatal period) and intrapartum period (skilled birth attendance, neonatal resuscitation, magnesium sulfate). While the postnatal intervention modelling has not been done such as small and sick newborn care.

So, reading this scoping review, I have some major comments on the paper and some minor comments to consider.

Major comments

1. Since, this was a PhD project of the first author, was there any set protocol in place and registration of the scoping review done. There are several systematic review registration sites, such as Joanna Briggs Institute, Campbell collaboration, Prospero, research registry. Prospective scoping review protocol will always help reviewers to check any deviation in the review from that of the original intent. There can be deviation, for which amendment of the registration can be done. If this was not done, the transparency of the review, a time comes into scrutiny. The process needs to be provided in the method section.

2. The researcher in the objective set out to find "how have mathematical modelling been used to evaluate the delivery of maternal and perinatal health intervention" and in your third specific objective the research explain it "to critically discuss the use of the common modelling approach". The paper discusses on "what modelling exercise, where, and why". I do not understand the scoping review entailing the use of the modelling in policy practice. I might have misunderstood, but please clarify that this undertaking was what component of different mathematical modelling.

3. In the search strategy in page 7, lines 156 to 159, you mentioned, that you originally limited your search for Lives Saving Tools, was that your original idea of the scoping review of focusing on the LiST intervention only. I think your scoping review encompasses all mathematically modelling or did you start with List and expanded your mathematically model to other. This is the reason; I am requesting for a prospective registration of the study.

4. In the result section, using table 3. a time series presentation on the different modelling being done over the years using different color codes would be very interesting. You present in the table 2, the years of publication, an elaboration will be interesting. Moreover, the year of publication to the different modeling done for timing of intervention and type of intervention will be of value for two important reasons. Global perinatal health expanded in the MDG era with the focus first on the essential care for mothers and newborns and then focusing on high-risk care. The different years when different modeling by interventions were done will be of value to the reader. Your supplementary table 2 has all the ingredients to it.

5. The main component of your work is the analysis of the different mathematical modeling i.e table 4, the explanation of the different model structure especially the linear and non-linear feedback, the deterministic vs stochastic model needs explanation, as the model output varies especially due to these two factors. I think a good table where different models used linear vs non-linear, deterministic vs stochastic is highly recommended.

Minor comment

Table 5 can be merged with table 2 or 3 or having a graphical presentation.

Some minor comments, please do not provide or explain the details of different intervention such a between lines 268 to 279.

Title is slight confusing, can you have " The mathematical models been used to evaluate the delivery of maternal and/or perinatal health interventions and their impact on health-related outcomes in sub96 Saharan Africa?”.

Reviewer #2: This is an interesting manuscript by Joseph and colleagues and can provide new literature on the different mathematical models used to project coverage and mortality.

I have a few comments in place

- In the abstract, can you make it sub-titles with introduction, method, result and discussion. The term such as "Models were most applied to estimate impact" needs to be quantified in the result section. It would be good to provide which software's were used in different models, if possible, in the abstract.

-Can you provide the list of different mathematical models in the introduction section of main text

-Can authors provide the policy and programmatic implication of this in the introduction section

-Why was this scoping review limited to Africa and not to Asia?

- Was English language literature only searched or in other languages such as Western African countries use other European language.

- I think it would be good to see the comparison of the estimates provided by different mathematical model, as it provides the variance.

6. PLOS authors have the option to publish the peer review history of their article (what does this mean?). If published, this will include your full peer review and any attached files.

Reviewer #1: No

Reviewer #2: No

---

## [Author Response · Author response to Decision Letter 0]

5 Aug 2024

We thank the reviewers and the editors for their time and the review of this manuscript. Please see our response to each of theses comments in the attached file.

---

## [Decision Letter · Decision Letter 1]

14 Nov 2024

Mathematical modelling to estimate the impact of maternal and perinatal healthcare services and interventions on health in sub-Saharan Africa: A scoping review

PONE-D-23-41936R1

Dear Dr. Collins,

We’re pleased to inform you that your manuscript has been judged scientifically suitable for publication and will be formally accepted for publication once it meets all outstanding technical requirements.

Kind regards,

Ashish KC

Academic Editor

PLOS ONE

Additional Editor Comments (optional):

Dear Dr. Collins

Based on the reviewers' comments and my assessment as editor. The paper reads well.

Reviewers' comments:

Reviewer's Responses to Questions

**Comments to the Author**

1. If the authors have adequately addressed your comments raised in a previous round of review and you feel that this manuscript is now acceptable for publication, you may indicate that here to bypass the “Comments to the Author” section, enter your conflict of interest statement in the “Confidential to Editor” section, and submit your "Accept" recommendation.

Reviewer #1: All comments have been addressed

2. Is the manuscript technically sound, and do the data support the conclusions?

Reviewer #1: Yes

3. Has the statistical analysis been performed appropriately and rigorously? 

Reviewer #1: Yes

4. Have the authors made all data underlying the findings in their manuscript fully available?

Reviewer #1: Yes

5. Is the manuscript presented in an intelligible fashion and written in standard English?

Reviewer #1: Yes

6. Review Comments to the Author

Reviewer #1: Thank you for the revision.

My comments have been addressed. I think the revision adding the year of publication to the different modeling done for timing of intervention and type of intervention will be of value for two important reasons. Global perinatal health expanded in the MDG era with the focus first on the essential care for mothers and newborns and then focusing on high-risk care has been well addressed.

7. PLOS authors have the option to publish the peer review history of their article (what does this mean?). If published, this will include your full peer review and any attached files.

Reviewer #1: No

---

## [Editor Report · Acceptance letter]

19 Nov 2024

PONE-D-23-41936R1 

PLOS ONE

Dear Dr. Collins, 

I'm pleased to inform you that your manuscript has been deemed suitable for publication in PLOS ONE. Congratulations! Your manuscript is now being handed over to our production team.

Kind regards, 

on behalf of

Dr. Ashish KC 

Academic Editor

PLOS ONE